# μXRF Mapping as a Powerful Technique for Investigating Metal Objects from the Archaeological Site of Ferento (Central Italy)

**DOI:** 10.3390/jimaging6070059

**Published:** 2020-06-30

**Authors:** Giuseppe Capobianco, Adriana Sferragatta, Luca Lanteri, Giorgia Agresti, Giuseppe Bonifazi, Silvia Serranti, Claudia Pelosi

**Affiliations:** 1Department of Chemical Engineering Materials & Environment, Sapienza, Rome University, Via Eudossiana 18, 00184 Rome, Italy; giuseppe.capobianco@uniroma1.it (G.C.); giuseppe.bonifazi@uniroma1.it (G.B.); silvia.serranti@uniroma1.it (S.S.); 2Department of Linguistic, Literary, Historical, Philosophical and Juridical Studies, University of Tuscia, Largo dell’Università, 01100 Viterbo, Italy; sfead@libero.it; 3Laboratory of Diagnostics and Materials Science, Department of Economics, Engineering, Society and Business Organization, University of Tuscia, Largo dell’Università, 01100 Viterbo, Italy; llanteri@unitus.it (L.L.); agresti@unitus.it (G.A.)

**Keywords:** archaeological metal objects, Tuscia region, μXRF mapping, bronze alloy, principal component analysis, multivariate curve resolution

## Abstract

This research concerns the application of micro X-ray fluorescence (µXRF) mapping to the investigation of a group of selected metal objects from the archaeological site of Ferento, a Roman and then medieval town in Central Italy. Specifically, attention was focused on two test pits, named IV and V, in which metal objects were found, mainly pertaining to the medieval period and never investigated before the present work from a compositional point of view. The potentiality of µXRF mapping was tested through a Bruker Tornado M4 equipped with an Rh tube, operating at 50 kV, 500 μA, and spot 25 μm obtained with polycapillary optics. Principal component analysis (PCA) and multivariate curve resolution (MCR) were used for processing the X-ray fluorescence spectra. The results showed that the investigated items are characterized by different compositions in terms of chemical elements. Three little wheels are made of lead, while the fibulae are made of copper-based alloys with varying amounts of tin, zinc, and lead. Only one ring is iron-based, and the other objects, namely a spatula and an applique, are also made of copper-based alloys, but with different relative amounts of the main elements. In two objects, traces of gold were found, suggesting the precious character of these pieces. MCR analysis was demonstrated to be particularly useful to confirm the presence of trace elements, such as gold, as it could differentiate the signals related to minor elements from those due to major chemical elements.

## 1. Introduction

This contribution concerns the application of micro X-ray fluorescence (μXRF) mapping to the investigation of a group of selected metal objects from the archaeological site of Ferento, a Roman and then medieval town in Central Italy. The main aim of this work was to test the potentiality of μXRF mapping for the characterization of archaeological metal alloys, which are particularly difficult to study due to the presence of oxidation patina and surface depositions that prevent the possibility of obtaining a strictly quantitative analysis.

Ferento was an ancient town, located approximately 8 km from Viterbo along the north-eastern direction, and one of the few urban settlements on the western side of the Tiber river (Figure 1) [1]. After the “social wars” (91–88 BC), Ferento was elevated to the rank of Roman Municipium, and thus a flourishing period began for the town; in the Julian–Claudian period, an intense urban planning program led to the construction of thermal baths, the Temple of Augustus in the Forum, and also a theater and an amphitheater. During the medieval period, the alternating historical events marked the slow decline of the town, culminating in its destruction between 1170 and 1172 by the neighboring town of Viterbo. The University of Tuscia started a relevant excavation campaign in 1994 that lasted for about 20 years. This research focuses on two test pits, named IV and V (Figure 2), in which several metal objects were found, including several dating to the medieval period [2]. In an archaeological context, these metal objects are relevant, especially for understanding aspects of the daily life of the settlement, such as building systems, craftsmanship, agriculture, military activity, and domestic life.

The metal objects were preliminary classified by archaeological specialists according to their use (i.e., objects for working, for building, for military equipment, for personal use, furnishings, harnesses for animals and other various pieces not attributable to the previous categories) and a database was created to house all of this information [2]. The metal objects from Ferento were never investigated before the present study and, in general, little attention is reserved for the characterization of archaeological medieval artifacts especially compared to Roman metallurgical artifacts. For this reason, the results obtained by the present study are relevant, especially for obtaining fundamental information about elemental compositions of the objects, and further, for better understanding the production of day-to-day items in the site of Ferento [3].

μXRF is certainly a valuable technique for the characterization of metal alloys, and remains in archaeological contests mainly because it is a non-invasive and non-destructive technique [4,5,6,7,8,9]. In fact, it provides useful information about alloy composition without the necessity of sampling, a step that is often not allowed for metal objects on display in museums [10,11,12,13]. Moreover, the mapping modality can provide knowledge of the whole composition of the surface layer, and can also afford information on the traces of possible surface decorations, even if corrosion patterns must be taken into account [14,15,16,17,18,19]. For this reason, principal component analysis (PCA) and multivariate curve resolution (MCR) were applied to the XRF spectra in order to investigate the similarities and differences between groups of objects (PCA) and to decompose the instrumental response for a mixture into the pure constitutions of each component (MCR).

## 2. Materials and Methods

### 2.1. Description of the Objects

The selected objects for μXRF mapping are described in Table 1. Each piece is identified with the stratigraphic unit that indicates the pit (first number, i.e., 4 or 5), the area of excavation, the inventory, and a short description of the object with its possible function and the date. The last category is based on a comparison with similar items found in other archaeological contexts and on the dating attributed by archaeologists to that specific stratigraphic unit.

It is difficult to backdate metal because the typology and the alloys remain the same for centuries. For example, the objects classified as ornamental jewellery (talismans or toys) in Table 1 had similar characteristics both in the Roman and in the medieval periods.

### 2.2. Cleaning of the Objects

The selected objects were cleaned before the XRF measurements by brush and a mixture of deionized water/ethyl alcohol for removing the bulk of the earth deposits.

Scalpels and micro-drilling were also applied, after the earth removal, to reduce the thickness of the oxidation patina and allow analysis of the alloys.

All cleaning operations were performed by a professional restorer, an expert in the conservation of metal objects from archaeological excavations, and were carefully checked under a stereomicroscope.

### 2.3. XRF Mapping

The investigated objects were examined using a Bruker^®^ (Billerica, MA, USA) M4 Tornado micro X-ray fluorescence (μXRF) benchtop spectrometer equipped with an Rh tube, operating at 50 kV, 500 μA, and spot 25 μm obtained with polycapillary optics, as well as with a XFlash^®^ detector providing an energy resolution better than 145 eV (at Mn Kα) and five filters [20,21]. Spectrum energy calibration was performed daily before each analysis by using zirconium (Zr) metal (Bruker^®^ calibration standard). The sensitivity of μXRF was determined by the excitation probability of the sample and the peak to background ratio. The background intensities were directly computed by the equipment (ESPRIT Bruker^®^ software). The sample chamber can be evacuated to 20 mbar and, therefore, light elements such as sodium can be measured [22]. Constant exciting energies of 50 kV and 500 μA were applied during the analysis. The set-up mapping acquisition parameters comprised a pixel size of 80 μm and an acquisition time, for each pixel, of 6 ms.

For XRF mapping, the objects were mounted on a thin layer (0.01 mm) of polyethylene, suspended in the chamber to reduce the noise of map acquisition, and pressed parallel to the stage table.

### 2.4. XRF Data Processing

The hypermap import and processing were preliminarily performed utilizing the ESPRIT M4 Tornado software and, after the extraction of raw data, statistical analysis was performed following a standard chemometric-based approach through the PLS_Toolbox (Version 8.6 Eigenvector Research, Inc., Manson, WA, USA) running inside MATLAB™ (Version 9.3). The XRF data were processed by PCA, which was used to decompose the “processed” spectral data into several principal components (PCs; i.e., linear combinations of the original spectral dataset) that embedded the spectral variations. The decomposition of the matrix *X* obtained by PCA is described by the following equation:(1)$$X=tp^{T}+E$$

The *T* scores matrix can represent the observations in the PCA space through the loadings of the matrix *P*, which contains information on the correlation between the variables [23]. According to this approach, a reduced set of factors is produced. Such a set can be used for discrimination, since it provides an accurate description of the entire dataset. The first few PCs resulting from PCA are generally utilized to analyze the common features among samples and their groupings: in fact, samples characterized by similar spectral signatures tend to aggregate in the score plot of the first two or three components. Spectra can thus be characterized either by the intensity at each peak in the KeV space, or by their score in the PCA space. Samples characterized by similar spectra, which belong to the same class of products, are grouped in the same region of the score plot, whereas samples characterized by different spectral features are clustered [24]. The singular value decomposition (SVD) algorithm was chosen and the fraction of samples (i.e., the rotation alpha) utilized in the SVD, for the scatter estimate, was 0.75 [25].

MCR was also applied to the μXRF data as a soft-modeling method, able to mathematically reduce the instrumental response of a mixture into the pure contributions of each component involved in the investigated system [26,27]. The strategy applied for MCR is to decompose a two-way data matrix *D* (*m* × *n*) into two matrices, namely, *C* (*m* × *k*) and *S^T^* (*k* × *n*), containing pure concentration profiles and pure spectra of the *k* species of the unknown mixture, respectively, according to Equation (2),
(2)$$D=CS^{T}+E$$
where *E (m × n*) is the error matrix containing the residual variation of the data. The analytical aim of MCR is to identify the main sources of variation, the components, and the profiles contributing to the raw measurement by using an additive model of linear contributions [28]. In XRF spectroscopy, for example, the bilinear model provides a simplified and interpretable information of the process data, under the premise that the multicomponent Beer’s law is valid. The components are represented by the dyads of vectors composed of a concentration profile in the process direction and a pure spectrum in the direction of the multivariate response. This model can be extended to more complex data arrangements (i.e., augmented data matrices) for which Equation (2) would still hold, such as for spectroscopic images or for second-order data complying with the requirements described above [29,30,31]. In our study, non-negative values for the concentration and spectra profiles were used as constraints, to implement alternating least squares (ALS) [32,33,34]. The advantage of the MCR approach is the fact that it provides meaningful models of the component profiles recognized in real instrumental responses [35]. The analyzed dataset was a hyperspectral XRF image, where the first two axes refer to the pixels in the x and y directions and the third to the number of spectral channels. In this case, the cubes should be unfolded into a sized data matrix (number of pixels *×* spectral channels) [28,36]. Before using any method of data analysis, pre-treatments are recommended to remove backgrounds or other variations unrelated to the components of interest, e.g., scattering. The aim of the pre-treatment strategy, adopted for the MCR approach, was to retain the properties of the original dataset [30]. In the present case study, baseline, smoothing, and normalizing treatments were used for background correction, thus retaining the positive characteristics of the instrumental signal. The constraints used for the MCR model were obtained by the standard calibration elements provided by Bruker^®^.

## 3. Results

The results of µXRF are reported as images, mapping XRF data in order to immediately highlight the main elements present in the examined objects, the distribution of each element, and the relative amounts of the detected elements. Images of the objects and the XRF elemental maps are shown in the Figure 3, Figure 4, Figure 5, Figure 6, Figure 7 and Figure 8.

In order to better highlight the compositional analogies between the objects, PCA was applied to the XRF spectra. The results of this analysis are reported both as score plots of the PCs and as loading plots that relate the variance with the XRF emission lines (Figure 9, Figure 10 and Figure 11).

MCR analysis was performed on the XRF spectra of the objects 5085_6, 5123_45, and 4021_12 (Figure 12, Figure 13 and Figure 14) in order to obtain a better differentiation in terms of the distribution between the signal related to the trace elements and the elements with major concentrations. In fact, in these objects, the XRF mapping revealed the presence of possible traces of gold (in 5085_6 and 5123_45) and of iron concentrated in two areas of the zoomorphic fibula 4021_12.

## 4. Discussion

### 4.1. The Three Ornamental Jewellery Objects (Talismans or Little Toys)

Despite the presence of patina on the surface of the pieces, although not particularly thick and partially removed during the cleaning operations, from the XRF maps shown in the Figure 3 and Figure 4, it can be seen that the three objects classified as ornamental jewellery (talismans or little toys), i.e., (4046_5, 4046_6, and 4155_1) are primarily made of lead. Moreover, if compared to each other, they appear to be homogenous in composition. They also contain a high percentage of Ca distributed on their entire surface (Figure 4). Elements such as Ca, Al, Si, P, K, Ti, Mn, and Fe can be associated with the soil where the objects were buried; in fact, they were present in all examined pieces with different amounts, probably due to the differences in soil composition of the excavation areas but also the cleaning operations that could not obtain a homogeneous thickness and distribution of the soil remnants on the surface of the objects. The high correlation between Ca and P in the maps of Figure 4 suggests the possible presence of bone and its degradation products nearby at the site of the excavation or of the clothing at the burial site. From the maps, is also possible to note a correlation between Fe an Mn (as well as oxides and other compounds of these elements) and between soil elements such as Si, Al, Ti, and K, which could be associated with silicates and aluminosilicates from the soil.

Zn was detected in some well-defined areas associated with Fe, Si, and K. This could suggest the possible presence of an original surface layer (decoration or other) only remaining in traces or of Zn corrosion product deposition at localized spots on the objects. It should be stressed that the three wheels have a significant thickness (approximately 1 cm) and are not flat. This causes distortions in the map, especially for elements that are concentrated in curved areas, such as Zn, and generates the blur effect, visible in the Figure 3, due to the distance from the focus point. It is one of the limitations in the mapping of 3D objects with this technique. To assess the presence or absence of zinc in the items, XRF spot measurements were carried out before mapping procedures.

PCA was applied to the XRF spectra (Figure 9) to better highlight the compositional similarities between the three little metal wheels. As is visible in the PCA score plot, low variability between the three examined objects is confirmed (Figure 9A). The created PCA model shows a variance cumulative of 97% with five principal components. The loading plot shows how the variance in the data depends mainly on Ca, Si, and Fe, which can be assumed to come from the soil in the excavation area (Figure 9B).

### 4.2. The Group of Fibulae, Rings and Earrings

Another group of objects, including fibulae, rings, and earrings, was investigated as a comparison, in order to discuss possible analogies and differences in terms of the detected elements (Figure 5 and Figure 6).

In general, it is possible to assess that, apart from the case of the object 4082_3, made of iron as its main element, all pieces are composed of copper-based alloys with high variability of Cu as well as of Sn/Zn/Pb (Figure 6).

In two objects, namely, 5085_6 (ornamental jewellery, earring) and 5123_45 (clothing accessory, fibula), both found in pit V, similar amounts of Cu, Zn, Sn, and Pb were detected (Figure 6). The typology of earring 5085_6 dates to the Roman period, but the fibula 5123_45 dates, instead, to the VI–VII century [2]. It may be that these two objects are dated back to the same period or that they are made of the same alloy due to the recycling of Roman metals to produce medieval pieces, as usually occurred in the Middle Ages [37,38,39,40,41].

In the fibula 5123_45, different values were obtained for lead and zinc in the ring and in the prong. The high content of Pb in the prong can be explained by the necessity of having a more malleable alloy to allow for folding of the piece to be inserted in the ring of the fibula. Finally, the two items 5123_45 and 5085_6 exhibited a further interesting detail, i.e., the presence of gold in traces, particularly concentrated in the left side of both rings. The presence of Au suggests the possible use of a golden finishing layer, now partially lost, and thus highlighting the precious character of these pieces. To better highlight and confirm the presence of gold, MCR analysis was performed on the XRF spectra (Figure 12 and Figure 13). The results obtained from this processing confirm the presence of a thin gold layer on the surface. In general, MCR can obtain a better differentiation in terms of the distribution between the signal related to the trace element (i.e., gold) and the elements with major concentrations (i.e., zinc). The MCR score image confirms the presence of zinc in the ring and of Pb traces on the surface of the prong.

Fibula 5000_27 is well-differentiated from the others due to the high content of tin (Figure 6) and to the presence of traces of zinc.

Fibulae 4025_4 and 4003_7 are the only two without tin; thus, they can be defined as brass objects. The probable use of expensive brass alloy suggests the relevance and precious character of these two pieces. The presence of a higher amount of As in 4025_4, with respect to those found in all of the other objects, suggests the possible intentional addition of this element with the aim of producing a stronger final item and with better casting behavior [42,43]. It must be said that the presence of As in metal alloys could also reflect the source of the mineral. Fibula 5050_4 may be classified as leaded red brass, i.e., a mixture of brass and bronze produced by scrap metal [40], but with different values of Zn, Sn, and Pb with respect to the previously discussed objects (i.e., 5123_45, and 5085_6).

The last examined object, within the group of fibulae, rings, and earrings, is that of 5079_5.

This item can be classified as red brass, containing Zn and Sn in similar amounts, but less than 6% of Pb (the normalized value of lead percentage gathered from the experimental spectra). Therefore, in this case, a probable brass master alloy was also used, combined with scrap bronze, without the addition of lead, to produce the piece. Clearly, due to the presence of a thin surface patina, a quantitative analysis is not possible, but a comparison between the different examined objects and the hypothesis on the original alloys used for the production of the pieces is allowed only in qualitative terms [11,13,15].

All items of the group were compared in the PCA score plot in order to better highlight the analogies and the differences (Figure 10). The obtained PCA model shows a variance cumulative of 99.97% with six principal components. The PCA score plot highlights the clear difference in the 4082_3 and 5000_27 items with respect to the other objects. In detail, the variance detected in PC1 allowed separation thanks to the presence of iron (i.e., ring 4082_3) for positive values and to copper for negative values (i.e., the other rings). PC3 shows how the variance was mainly due to zinc. The combination of PC1 and PC3 allowed the separation of sample 5000_27 from the other rings. The other items exhibited similar compositions, with the variance due to the presence of minor elements detected in the spectra.

For this group of objects, a correlation was also observed between Ca and P (see maps in Figure 6), suggesting the possible presence of compounds associated with bone or clothing in the burial site. Unfortunately, no information was supplied concerning the excavation conditions or soil composition, and so we can only hypothesize about them.

Lastly, a correlation may have been found for soil elements (i.e., Si, Al, K, and Ti) constituting silicates and aluminosilicates.

### 4.3. The Group of the Zoomorphic Fibula, the Little Spatula and the Furnishing Item

The last examined group of objects comprised three items: one zoomorphic fibula, a little spatula for cosmetics, and a furnishing piece (probably an applique, Figure 7 and Figure 8).

The choice of these objects was made in order to examine the alloy typology and to compare it to that found for the other investigated objects in the group of fibula/earring/ring (see Section 4.2). The three items are clearly different from each other, having various functions and use, but they are all constituted by copper-based metal alloys. The composition of the zoomorphic fibula is different from those of all other examined fibulae in terms of Zn and Sn. Fibula 4021_12 is leaded bronze, Sn-enriched, as visible in the elemental maps (Figure 8). This zoomorphic fibula is analogous to another piece found in the excavation of the medieval town of Winchester in England dated back to the XIII–XIV century [44]. In fact, its composition shows similarities with late mediaeval objects of Northern European origin [37,45].

Sequentially, PCA was applied also to the XRF spectra of this last group of investigated objects and the results are shown in Figure 11. The PCA model shows a variance cumulative of 99.81% with three principal components. The PCA score plot highlights three clouds, corresponding to the three examined objects. The PC1 loading plot shows how the variance was mainly related to Cu for positive values and to iron for negative value, while the PC2 loading plot shows how the variance was mainly related to Zn for positive values and to Pb for negative values. Sample 4021_12 was mainly influenced by negative values of PC1, whereas sample 4003_9 was mainly influenced by positive variances in PC1. Finally, sample 5123_27 was influenced by positive values of PC2. In this case, the variability of 5123_27 was also due to higher Zn with respect to the other two objects. The fibula 4021_12 does not have a prong, but the presence of Fe, concentrated in the joint between the heads of the birds and in the area where the prong stokes the fibula, clearly indicates that the prong was made of iron or an Fe-based alloy (see Figure 11). Fe is highly correlated to Mn, as is visible in the maps of these two elements for the zoomorphic fibula (Figure 8, maps of Fe and Mn).

To better highlight the distribution in variance of the detected elements, MCR analysis was performed on the XRF spectra of sample 4021_12. The results highlight the difference between the bulk of the object (Comp. 2) and the parts rich in iron (Comp. 1) (Figure 14). The other element detected (i.e., Sn in Comp. 3 and lead in Comp. 4) were well distributed in the analyzed sample (Figure 14). The presence, in Ferento, of this zoomorphic fibula is probably linked to the passage of pilgrims along the Via Francigena towards Rome [2].

The applique (item 4003_9) contains copper, tin, and lead, without Sn, and with the highest Cu content between the three examined objects. The use of brass for a furniture element suggests a precious and expensive object, as previously discussed. Lastly, the little spatula used for cosmetics (5123_27) can be classified as red brass, according to the percentages of Zn, Sn, and Pb. Lead is present in low concentrations, not visible in the map shown in Figure 8, due to the much higher content of this element in the applique and, above all, in the zoomorphic fibula.

A high correlation was observed for Ca and P in the maps of Figure 8, also suggesting the possible presence of bone and its degradation products or clothing in the burial site.

The correlation between Si, Al, and K indicates the presence of silicates and aluminosilicates in the soil. Lastly, arsenic (As) was mapped for the three pieces (with more evidence in the applique), due to its homogeneous distribution on the item. It can be supposed that it was contained in the source minerals used for the production of the object [38].

## 5. Conclusions

The present work proposes a suitable approach for the interpretation of XRF hyperspectral data arrays obtained by μXRF analyses of archaeological metal artifacts from the site of Ferento. These objects were never investigated before the present research and, for the first time, micro-compositional analysis was used to study said metal objects.

The XRF imaging technique, in combination with a multivariate methodology, proved to be a powerful and efficient tool to obtain valuable information from a large set of X-ray mapping spectra, definitely important for a straightforward interpretation of complex matrices. The MCR technique, in combination with the study of the spectral average profile, was useful to discriminate elements in traces (such as gold) from elements with major concentrations (such as zinc) with XRF signals at the same or similar energies (expressed in KeV) as the minor or trace elements.

Through PCA, differences between the analyzed objects in terms of elemental composition could be easily detected. The advantage of this approach can also be found in the evaluation of all data cubes in one shot without losing information on the variability of the data: this occurs when the analysis concerns only the mean spectra. The established significant correlations are useful for historical studies and to better explain the temporal evolution of the archaeological site. In comparison to the classical mapping technique, this chemometric methodology can obtain more information from the content-fast and non-expensive investigative procedure, which should always be applied in the examination of μXRF maps of samples with unknown composition, avoiding the loss of global information connected to an incomplete processing of hyperspectral data.

## Figures and Tables

**Figure 1 jimaging-06-00059-f001:**
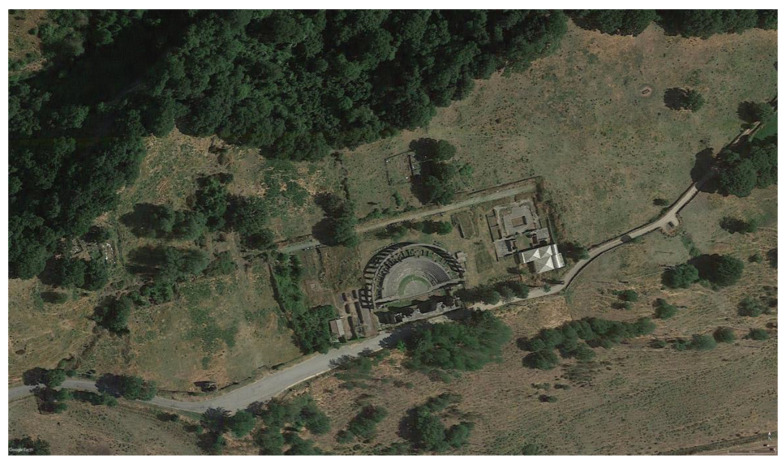
Aerial view of the archaeological site of Ferento with the Roman theater well-visible in the center of the image.

**Figure 2 jimaging-06-00059-f002:**
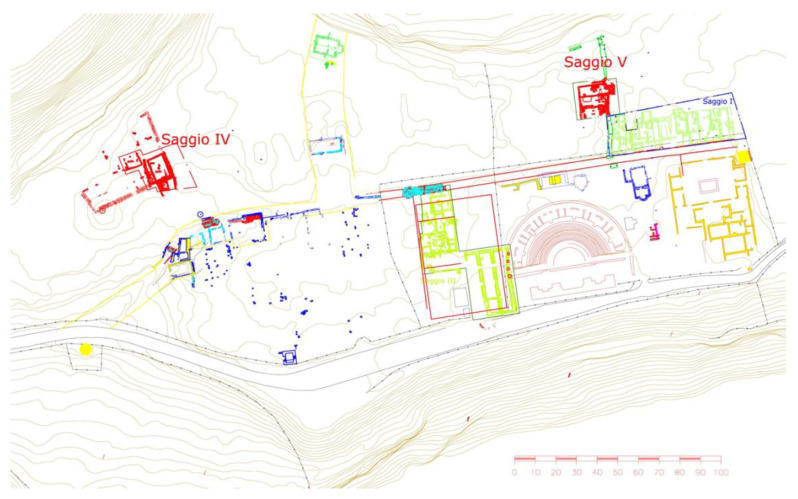
Planimetry of the excavation areas with the test pits (named IV and V) from which the examined objects were recovered.

**Figure 3 jimaging-06-00059-f003:**
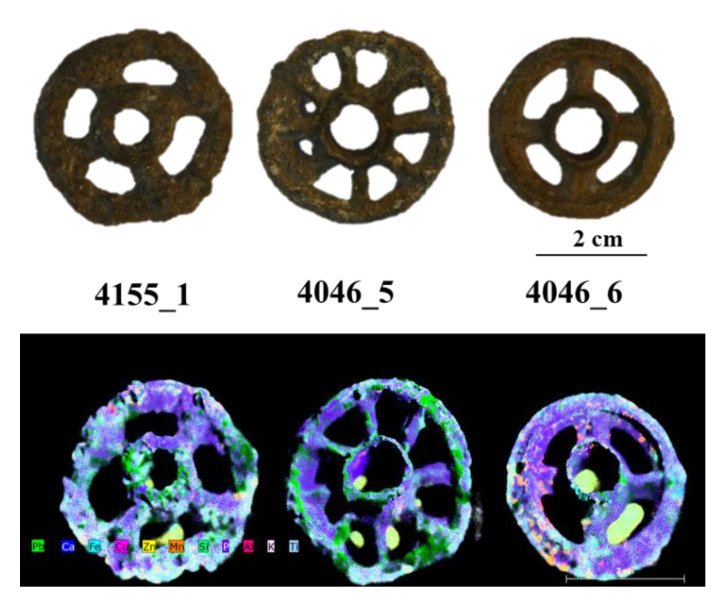
Photographs of the three ornamental jewellery objects and the XRF mapping of the detected elements.

**Figure 4 jimaging-06-00059-f004:**
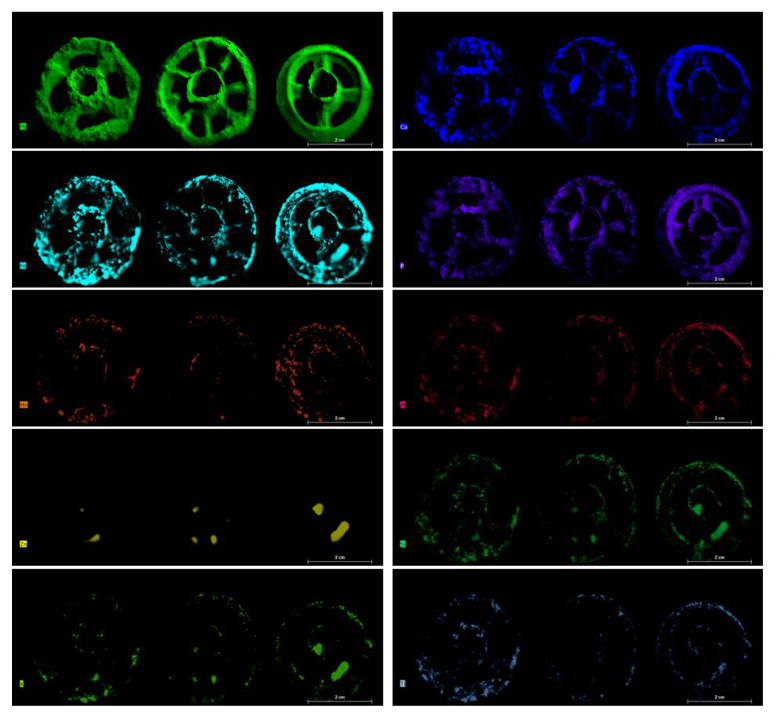
Elemental maps for the detected elements of the three ornamental jewellery items. The Cu map is not shown due to the extremely low content of this element.

**Figure 5 jimaging-06-00059-f005:**
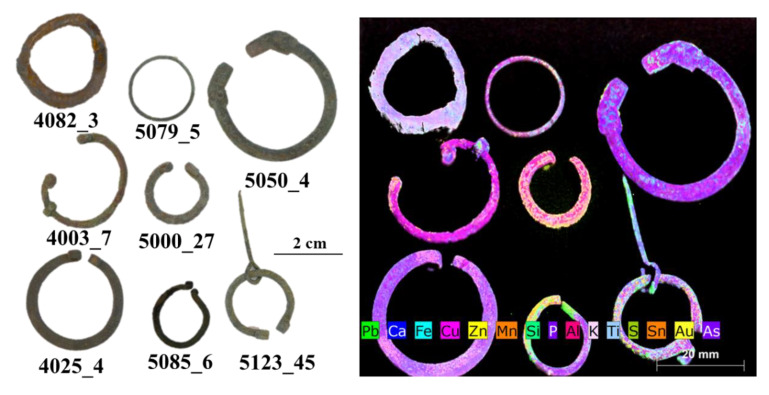
Photographs of the objects of the fibula/ring/earring group and the XRF mapping of the detected elements.

**Figure 6 jimaging-06-00059-f006:**
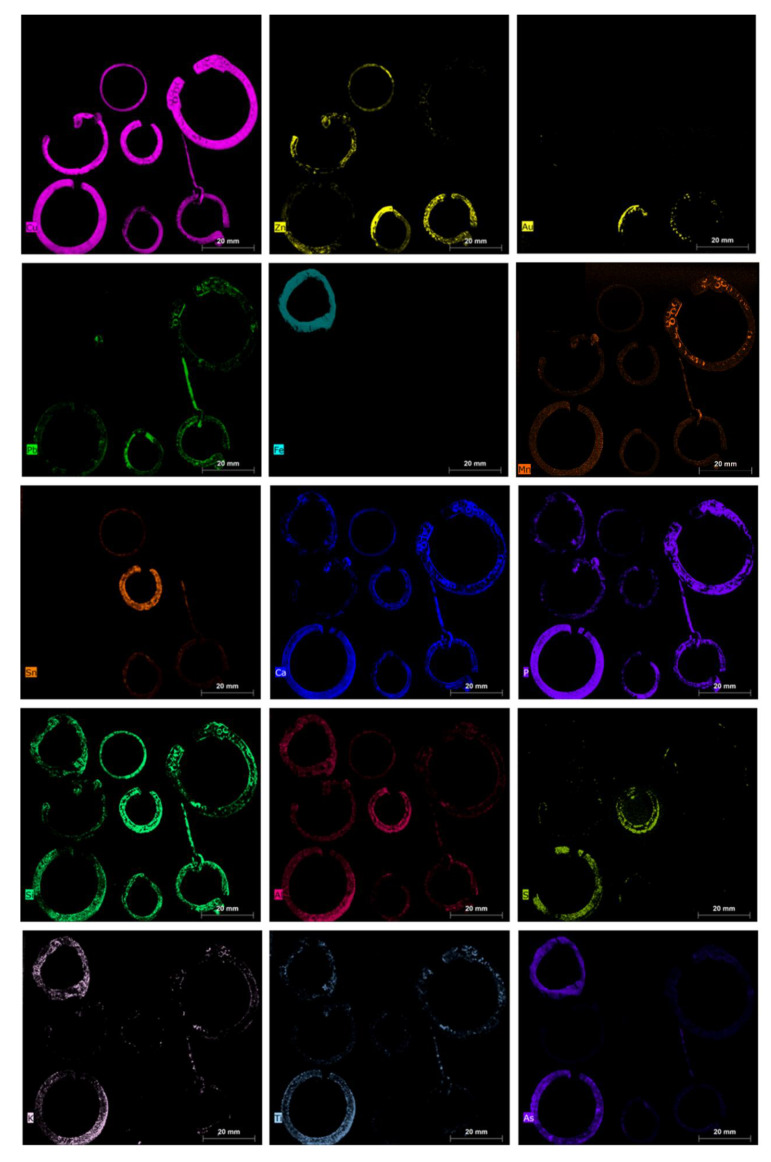
Elemental maps for all detected elements of fibula/ring/earring group.

**Figure 7 jimaging-06-00059-f007:**
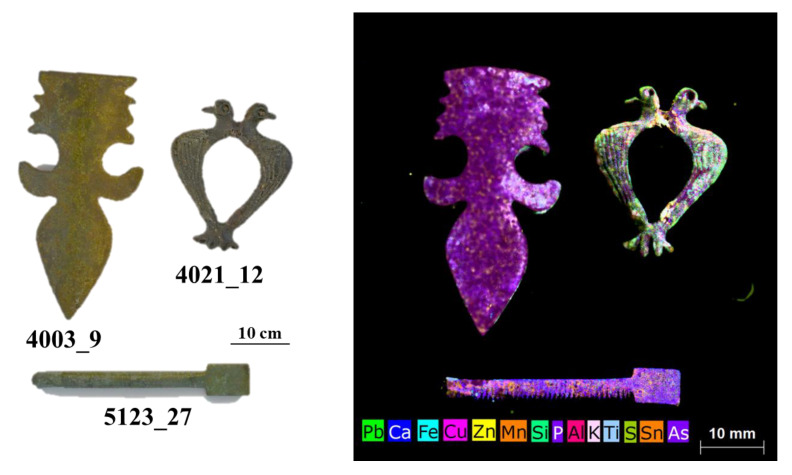
Photographs of the examined objects (i.e., applique, zoomorphic fibula, and spatula) and the XRF mapping of the detected elements.

**Figure 8 jimaging-06-00059-f008:**
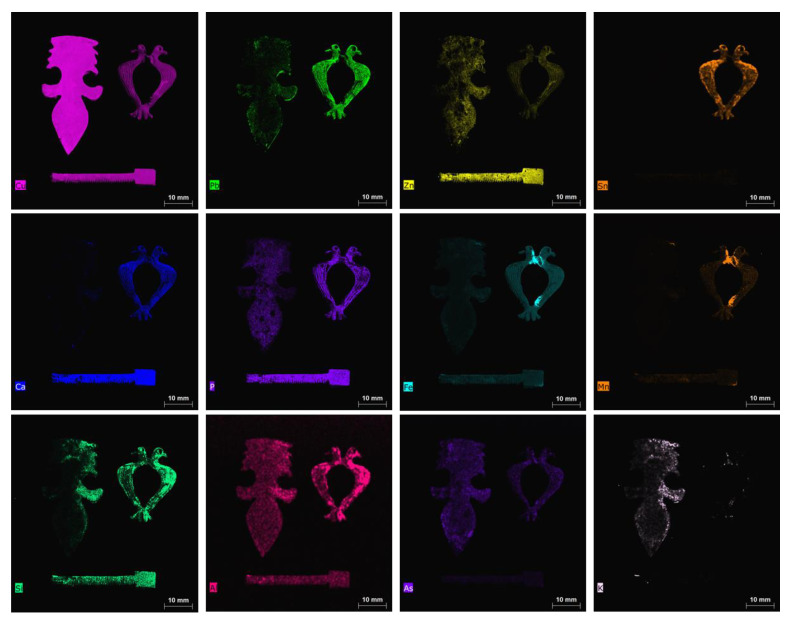
Elemental maps for the detected elements of applique, zoomorphic fibula, and spatula. The maps of S and Ti are not shown due to the extremely low contents of these two elements.

**Figure 9 jimaging-06-00059-f009:**
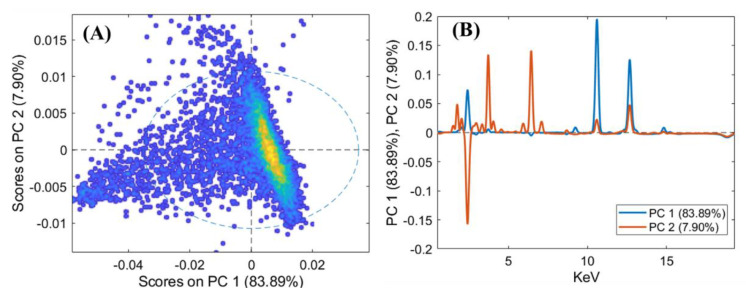
Principal component analysis (PCA) score plot of the XRF spectra (**A**) and the loadings of PC1 and PC2 (**B**) for the three lead wheels (i.e., ornamental jewellery, talismans, or little toys).

**Figure 10 jimaging-06-00059-f010:**
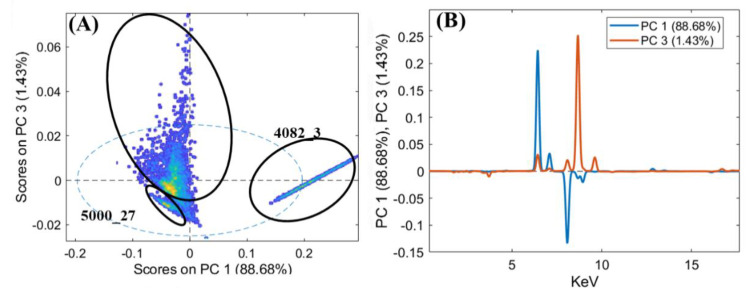
PCA score plot of the XRF spectra (**A**) and the loadings of PC1 and PC3 (**B**) for the fibulae (4003_7, 4025_4, 5000_27, and 5050_4), earrings (5079_5 and 5085_6), and ring (4082_3).

**Figure 11 jimaging-06-00059-f011:**
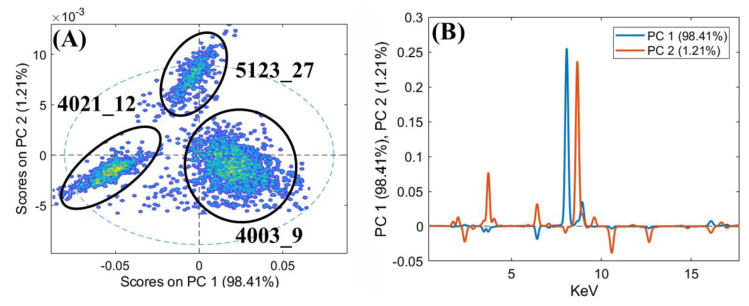
PCA score plot of the XRF spectra (**A**) and the loadings of PC1 and PC2 (**B**) for the zoomorphic fibula (4021_12), the applique (4003_9), and the spatula (5124_27).

**Figure 12 jimaging-06-00059-f012:**
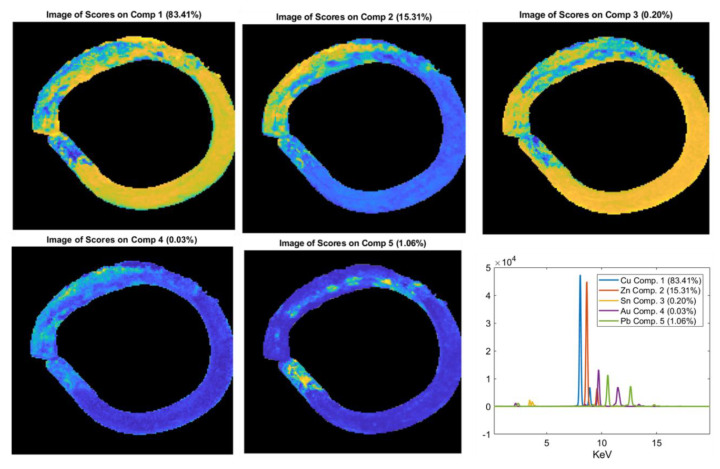
Multivariate curve resolution (MCR) analysis on the XRF spectra of the object 5085_6. Cu and Sn are uniformly distributed in the alloy (Comp. 1 and Comp. 3), while Zn (Comp. 2) is concentrated in a defined area. Traces of gold are confirmed (Comp. 4), and in a little area of the surface, a high concentration of Pb is obtained (Comp. 5).

**Figure 13 jimaging-06-00059-f013:**
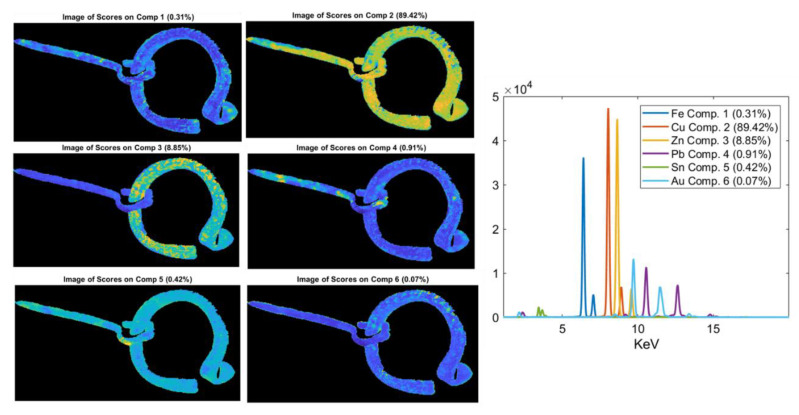
MCR analysis on the fibula 5123_45, the only one having the prong. Zn is uniformly distributed on the object’s surface (Comp. 3), while traces of gold are confirmed (Comp. 6), and a high Pb content is visible in the prong (Comp. 4).

**Figure 14 jimaging-06-00059-f014:**
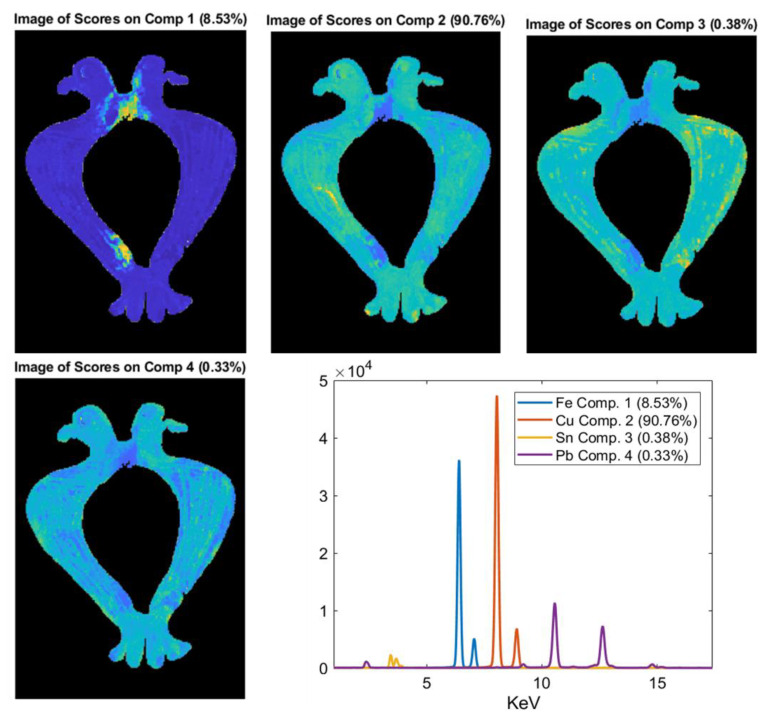
MCR analysis on the XRF spectra of the zoomorphic fibula 4021_12. This analysis highlights the difference in composition between the copper-based alloy (Comp. 2) and the parts rich of iron (Comp. 1).

**Table 1 jimaging-06-00059-t001:** Examined objects, according to the classification made by archaeologists [2].

Stratigraphic Unit	Area	Inv. Nr.	Description	Typological Dating of the Item	Dating of the Stratum
4003	L 8 III d	4003_7	Clothing accessory, fibula	VI–VII	XI–XII
4003	L 8 III a	4003_9	Applique, furnishing object	—	XI-XII
4021	L 7 II a	4021_12	Clothing accessory, zoomorphic fibula	XIII–XIV	Late Middle Ages
4025	L 8 III b	4025_4	Clothing accessory, fibula	VI–VII, and early XII	XI–XII
4046	L 7 II a	4046_5	Ornamental jewellery, talisman or toy	From the Roman to the Medieval period	XI–XII
4046	L 8 III d -L 8 IV c	4046_6	Ornamental jewellery, talisman or toy	From the Roman to the Medieval period	XI–XII
4082	L 8 III c	4082_3	Ring	Middle Ages	XII
4155	L 7 I a	4155_1	Ornamental jewellery, talisman or toy	From the Roman to the Medieval period	XII–XIII
5000	L 18 IV c	5000_27	Clothing accessory, fibula	VI–VII, and early XII	Humus and XII–XIII
5050	L 18 I b	5050_4	Clothing accessory, fibula	VI–VII	Middle Ages
5079	L 18 I b	5079_5	Ornamental jewellery, probable earring	—	XII–XIII
5085	L 18 I d	5085_6	Ornamental jewellery, earring	Roman period	XI–XII
5123	L 18 IV d	5123_27	Spatula for cosmetics	Late III century	XI–XIII
5123	L 18 IV d	5123_45	Clothing accessory, fibula	VI–VII	XI–XIII

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
