# Peer review of "μXRF Mapping as a Powerful Technique for Investigating Metal Objects from the Archaeological Site of Ferento (Central Italy)"

_2313-433X, 2020, doi:10.3390/jimaging6070059_

Round 1

Reviewer 1 Report

This paper explores μXRF mapping as a technique to analyze metal artifacts recovered from Ferento (a Roman and then medieval town). Scientifically, I believe that the article is quite unique and presents a very interesting approach to the compositional study of metallurgical artifacts. However, the article needs major language revisions and it also needs some more archaeological grounding. Based on your definitions for the recommendations, I am selecting minor revisions, but please note that I believe the authors need to dramatically improve the text. I have provided some comments and corrections throughout the text, but they are just some of the errors. If the authors are willing to significantly revise the manuscript, then I believe that article should be accepted.

Accept after minor revision (corrections to minor methodological errors and text editing)

Introduction

Page 1: You say that Ferento was an important town in Roman times and during the Middle Ages, but you never explain why it was important (other than to say it was one of the few urban settlements in the area). Was it a trading town? A little more context would be useful.

Page 2: Rephrase from “After the Roman phase, the archaeological site of Ferento played an important role even in the Middle Ages, until the end of the 12th century when it was definitively destroyed by Viterbo.” to “The archaeological site of Ferento played an important role in the region until the end of the 12th century when it was destroyed by the neighboring town of Viterbo.”

Page 2: Rephrase to “This contribution focuses on two test pits, named IV and V (Fig. 2), in which several metal objects were found, including several dating to the medieval period [2].”

Page 2-3: Rephrase from “The metal objects have been preliminary classified by archaeologists according to the use, i.e. objects for working, for building, for military equipment, for personal use, furnishings, harnesses for animals and other various pieces not attributable to the previous categories, creating a database with all items [2].” To “The metal objects have been preliminary classified by archaeological specialists according to use (i.e., objects for working, for building, for military equipment, for personal use, furnishings, harnesses for animals and other various pieces not attributable to the previous categories) and a database has been created to house all of this information.”

Page 3: Change “were never been” to “have never been”

Page 3: Change “and in general few attention is reserved to” to “and in general, little attention is reserved for”

Page 3: Mediaeval – change to medieval

Page 3: Change “in respect to Roman one, for example” to “especially compared to Roman metallurgical artifacts.”

Page 3: Change “In fact, it allows to obtain useful information about alloys composition without necessity of sampling, being this step often not allowed as in the present case and generally for metal objects exposed in the museums [10-13].” To “In fact, it provides useful information about alloy composition without the necessity of sampling, a step that is often not allowed for metal objects on display in museums [10-13].”

Page 3: Is the last paragraph in the introduction necessary? It reads like a table of contents for the paper and it is not useful or informative for the reader. However, if this is a journal formatting requirement or stylistic preference, by all means, keep it.

Materials and Methods

Page 3: Change “dating” to “date”

Page 3: Change “This last” to “The last category”

Page 3: Change “Metal objects are difficult to be dated back” to “Metal objects are difficult to date”

Page 3: Change “professional restorer, expert” to “professional restorator, an expert”

Page 4: Change “were adopted for acquisition” to “were applied during the analysis.”

Page 4: Change “preliminary” to “preliminarily”

Page 4: Change “decompose” to “separate or break up”

Page 4: Change “as soft-modelling” to “as a soft-modelling”

Page 4: Change “decompose” to “reduce”

Results

Page 5: Change “micro X-ray fluorescence” to “μXRF”

Discussion

Page 11: Change “are made of lead, as main element.” To “are primarily made of lead.”

Page 11: Change “analogies” to “similarities”

Page 11: Change “analogies” to “similarities”

Page 12: “Ring 4082_3 was included, following the request by archaeologists, being found in a tomb and so its dating can be assessed to the XII century.” This is problematic as you have stated that all materials presented in the article come from two test pits, but now for the first time you are mentioning an object from a tomb. Either mention the context of the ring earlier or remove the object from the manuscript.

Page 12: Change “The earring’s typology of 5085_6 can be referred to the Roman period,” to “The typology of earring 5085_6 dates to the Roman period, but”

Page 12: Mediaeval – change to medieval                 

Page 12: Change “enables to confirm” to “confirms”

Page 13: Change “The three items are clearly different each other” to “The three items are clearly different from each other,”

Page 13: Delete “Let’s start with the discussion about the zoomorphoic fibula.” Change “The composition of the zoomorphic fibula is…”

Page 13: Mediaeval – change to medieval                 

Page 13: “Lastly, the little spatula used for cosmetics and classified with the nr. 5123_27.” This is a fragment and not a complete sentence – rework.

Conclusions

Page 13: Delete “objects, in particular of a selected group of metal items” add “archaeological metal artifacts from the site of Ferento.”

Page 13: Delete “a micro-analytical campaign was started for” change to “and for the first time, micro compositional analysis was used to study the metal objects from Ferento.”

Page 13: trace elements (like gold)

Page 13: There has been no case to support why the presence of gold is importance for this medieval town. Stress the importance here in the conclusion and how that links back to your intro when you talk about the importance of the town in the Roman and Medieval periods.

Funding

Page 14: Change “founded” to “funded”

Figures

Figure 2: Change to “Planimetry of the excavation areas with the test pits named IV and V from which the examined objects were recovered.

Figure 9: Change the caption to read “loadings of PC1 and PC2”

Figure 10: Change the caption to resemble Figure 11’s caption in which the objects are clearly labeled in PC biplot and the caption.

Figure 14: Spectrum figure appears distorted and stretched – fix before final publication

Author Response

Dear Reviewer,

We would like to thank you for your careful revision work.

The point-by-point response is supplied in the attachment.

Thanks

Claudia Pelosi (corresponding author)

Reviewer 2 Report

The paper by Capobianco et al presents an interesting application of μ-XRF mapping and XRF data manipulation that pertains to the analysis of archaeological metal items.

Despite its merits, to the reviewer’s opinion the text needs improvement prior to its publication.

First of all, more attention must be paid on the fact the analyzed items come from excavation contexts and are corroded. Despite the partial removal of corrosion layers described in Methodology (p. 3, section 2.1), analytical data reveal that the “cleaned” items bear still soil contamination. On this basis, discussions pertaining to presumable “alloy compositions” etc should be eliminated or at least revised accordingly. In order for the authors to get an idea of the “real” alloy composition, they should have conducted XRF analysis on small areas from which the surface patina layers would have been removed completely. I do understand the limitations imposed by the archaeological character of the items, yet to my view this is the only way to estimate alloy composition by surface XRF analysis.

I also suggest that the data manipulation processes (PCA and MRC) be a bit more elaborated in the Methodological part of the paper.

Finally, the text needs a substantial improvement as regards English.

A list of specific comments/suggestions follows.

Page 3, lines 16- (“In section 2…): I am not accustomed with such lengthy descriptions of a paper’s subsections; authors may like to consider omitting (or revising) this paragraph.

Page 5, line 12, “Bruker®time 0 h”: please elaborate/revise.

Pages 5 & 6, figures 3 & 4: there are some zinc- and iron-rich spots that are recorder on the μ-XRF mappings, which in fact are not present on the items’ photographs (fig. 3-upper part). Can you please elaborate on these spots? (see also another relevant comment below)

Page 11, discussion, lines 7-8 “with different amounts due to the differences in soil composition of the excavation areas”: is it indeed possible to compare soil contamination compositions between the various items? If yes, this implies considerable remnants of soil which seems rather improbable given the cleaning pre-treatment. Maybe the variations on the earth elements intensities correspond to thickness variations and/or degree of soiling removal upon scalpel/mechanical cleaning.

Page 11, discussion, lines 8-9, “Zn has been detected in some well-defined areas where it is associated to Fe. This…now remained in traces”: upon inspection of the wheels’ photos (figure 3-up) I suspect that these Zn/Fe-rich phases do not correspond to the analyzed items. See another relevant comment above and please elaborate.

Page 12, line 4, “exhibit similar composition in terms of main elements of the alloy”: due to the limitations imposed by the presence of surface oxidation layers I would suggest that the authors be more prudent when commenting on “alloy compositions”.

Page 12, lines 40-41, “Clearly, due to the presence of a thin surface oxidation patina, a quantitative analysis is not possible…”: please consider the aforementioned limitations regarding analysis of corroded metallic items. Authors should provide a more sound relevant discussion (maybe in their methodological part?) and cite pertinent literature (that abounds).

Page 13, line 21, “fibula, clearly indicates that the prong was made of Fe rich alloy”: do authors indeed infer employment of a Fe-rich alloy or a Fe-based one?

Page 13, lines 31-32, “This a typical alloy categorized in the group of sheets”: unclear meaning, please revise.

Page 13, Conclusions, lines 11-12, “is a relevant finding because it demonstrates the importance of the town in the mediaeval period.”: to the reviewer’s opinion, the presence of gold traces on a single or a couple of items is by no means an indication of a wealthy society.

Indicative English errors: page 3, first line ‘…from Ferento were never been investigated…” / p. 3, line 3 “Roman one” (instead of Roman ones) / page 10, line 5: “High Pb are visible in the prong” / page 11, line 4: “if compared each other” / page 13, line 3 “…three items are clearly different each other...” etc.

Author Response

(The authors gave the same response as above.)

Reviewer 3 Report

General Comments

XRF is a sensitive technique for the investigation of metallic materials.The contribution deals with the application of micro-XRF mapping along with chemometric methodologies for the analysis of corroded archaeological metallic objects excavated from the site of Ferento. The multivariate statistical approaches have enabled the separation of chemical groupings and individual components of the materials investigated. The application of fast, non-destructive and non-invasive methodology is crucial and highly recommendable in the context of preserving the material integrity of the precious cultural heritage objects that are unique and irreplaceable in their nature. Great emphasis is often given to that aspect of analytical techniques from the conservation and ethical perspectives. This work is, thus, a useful contribution in that context. 

The study is well planned, the method described sufficiently and convincing interpretations provided based on the experimental data gathered. However, there is room for improving the interpretation provided in some of the cases addressed. The use of complementary techniques could further improve the interpretation and shed more light on the structure and composition of the archaeological objects studied. That could be addressed in future investigations of similar objects.

Restructuring the short paragraphs in the current manuscript is also recommended.

Specific Comments and Suggestions

The manuscript sent for the review has no numbering for the text lines. I have tried to give them numbers for each sub-sections to facilitate the reference to the specific comments.

Abstract

Line 9: … compositions

Introduction

Line 5: … oxidation patina and surface depositions …..

Line 22: .. from Frento have never been……

Line 26: … for better understanding of the ……

Line 26: Regarding the objective\relevance of the experimental results mentioned here, does that mean the site of excavation has indications of metal production\processing or the objects found there are supposedly made elsewhere and found on the site? There is no account of slag materials, ceramic, crucible fragments, etc. described in the text.

Line 30: … without the necessity of ….

Line 31: …. The metal objects exhibited or displayed …

Line 32: … … modality can provide knowledge …

Line 35: …. on the XRF spectra …

Line 38-39: Consider excluding this paragraph as description of the structure in the article may not be that much necessary.

Materials and Methods

Line 2-9: Can be one paragraph

Line 10-15: Can be one paragraph

Line 17: … were examined using ….

Line 20: 145 eV resolution (at Mn Ka?)

Line 31: Table 1 caption … objects, according to the classification ….

Line 43: Varimax rotation used in the PCA analysis?

Results

Line 5: Figure 3 - Zn is found in localized spots\areas? What could be the possible reason? Deposition of corrosion products mainly composed of Zn compounds? Any correlation with mapping of sulphur, chlorine, oxygen or carbon, to get a hint about possible corrosion product formation? The elemental mapping beyond deciphering the possible composition of the bulk metallic material (more realistically the surface depositions that could be indicators of the metal composition), it can help identify the transformation of the metals when exposed to the external environment factors including depositional conditions. This important information could facilitate the well-informed conservation interventions to be carried out on the metallic objects.

Line 7: Figure 4 - The simultaneous detection of Ca and P (high degree of correlation) could possibly imply calcium phosphate-rich soil at the deposition site from which the ornamental jewelry items were recovered. Could that mean they were likely in close proximity with bone material? Is the specific excavation site for these objects recovery associated with burial ground from the archaeological interpretations? There are also indicators of the surface to to rich in soil matter (Fe, Al, Mn, probably same with Si- implying aluminosilcates, as well as oxides of iron and manganese). See also the comment for the localized Zn clumps identified from the mapping – corrosions emanating from the ornamental jewelry or something else?

Line 13: Figure 8 - The main target in this investigation could be the metallic components. However, information on corrosion products, surface depositions, etc. are important to describe the burial condition, pre-deposition alterations, state of conservation and the like. It is, thus, great to have the mapping of relevant elements (Cl, P, S, Cl, aluminosilicates, Fe, etc.). The case of Ca and P correlation, mentioned earlier, is one case in point.

Line 17: Caption, figure 9 - PC3? Does it mean PC2?

Line 26: Caption Figure 12 - Cu and Sn (Zn?) More concentration at the surface is detected for Zn at certain section of the object. Does Comp 2 represent Zn?

Line 30: Caption Figure 13 – Traces of Au (Comp 2?). I think comp 2 is for Cu mapping. Au might be represented by comp 6. What do you think? Comp 4 may also represent Sn due to the relative contrast in the concentration to the tip of the prong and at the joining site. Comp 5 for lead then?

Line 33 -34: Comp 1 (you mean comp 2?) and similarly, comp 2 (comp 1?). Looks like comp 1 is Fe, and comp 2, Cu.

Discussion

Line 5: Correlation of Ca with P can be mentioned here. Bone and its degradation products in nearby at the site of the find? Could possibly refer to part of a clothing at burial site?

Line 10: Could it also be Zn corrosion product deposition at localized spots\areas on the objects? Closer examination with methodologies that combine high resolution imaging with elemental analysis (like SEM-EDS) could shed more light on microstructure and composition of the product formed and\or intact metallic parts.

Line 16-21: Can be one paragraph.

Line 20: Ring 5085_6 is darker in colour compared with the brighter 5123_45, though both contain lead. The lighter tone in the case of later could possibly be due lead white, lead basic carbonates. Any sulfur associated with the former?

Line 25-27: Care needs to be taken in the interpretation of the experimental data in this case. Surface composition may not necessarily reflect the bulk composition of the unaltered metal. Structural information from high resolution imaging coupled with elemental analysis can shed more light on the characterization and, thereby, facilitate the interpretation.

Line 31: Which is relatively present at higher concentration? Zn or Sn? The mapping (Figure 6) seems to show more of the Zn. Can it also be regarded as brass then? Again there is need to consider the deposition and dissolution phenomena at the surface (with respect to each element) that could make the correlation with the unaltered metal core not straightforward.

Line 44: … from scrap bronze… How about minor\traces of elements from the mineral source used in the smelting of the relevant metal? Can it be totally ruled out?

Line 48: … producing a stronger final item …. Could it also simply reflect the source mineral in some cases?

Line 71-80: Could be in one paragraph?

Line 80: Analogies with other pieces - If possible, nice to have a short description of the other objects, used for the comparison, in terms of their compositions - just for the sake of clarity.

Line 102: Zn, S, Pb in the spatula – Sn is significantly very trace as can be discerned from the mental mappings displayed. The metal is not leaded as the element is barely detected from the mapping images (Pb is heavy element, and XRF very sensitive for its detection).

Conclusions

Line 7: … from large set of X-ray mapping spectra

Line 8-10: Not clear, may need rewriting the sentence.

Line 18: …., this chemometric methodology allows…..

Line 19. Better to use short sentences her. … content of the XRF images. It represents an efficient, fast, …..

Author Response

(The authors gave the same response as above.)

Round 2

Reviewer 2 Report

Dear Authors,

Your paper has been significantly improved. Therefore I suggest it is accepted in its current form. I shall only note that in the references list the paper by Azzouz and Tauler 2008 appears twice (references 33 &34).